# Towards efficient representation identification in supervised learning

**Kartik Ahuja**[*]                                                    KARTIK.AHUJA@MILA.QUEBEC
*Mila - Quebec AI Institute, Université de Montréal.*

**Divyat Mahajan**[*]                                                  DIVYAT.MAHAJAN@MILA.QUEBEC
*Mila - Quebec AI Institute, Université de Montréal.*

**Vasilis Syrgkanis**                                                  VASY@MICROSOFT.COM
*Microsoft Research, New England*

**Ioannis Mitliagkas**                                                 IOANNIS@MILA.QUEBEC
*Mila - Quebec AI Institute, Université de Montréal.*

**Editors:** Bernhard Schölkopf, Caroline Uhler and Kun Zhang

## Abstract

Humans have a remarkable ability to disentangle complex sensory inputs (e.g., image, text) into simple factors of variation (e.g., shape, color) without much supervision. This ability has inspired many works that attempt to solve the following question: how do we invert the data generation process to extract those factors with minimal or no supervision? Several works in the literature on non-linear independent component analysis have established this negative result; without some knowledge of the data generation process or appropriate inductive biases, it is impossible to perform this inversion. In recent years, a lot of progress has been made on disentanglement under structural assumptions, e.g., when we have access to auxiliary information that makes the factors of variation conditionally independent. However, existing work requires a lot of auxiliary information, e.g., in supervised classification, it prescribes that the number of label classes should be at least equal to the total dimension of all factors of variation. In this work, we depart from these assumptions and ask: a) How can we get disentanglement when the auxiliary information does not provide conditional independence over the factors of variation? b) Can we reduce the amount of auxiliary information required for disentanglement? For a class of models where auxiliary information does not ensure conditional independence, we show theoretically and experimentally that disentanglement (to a large extent) is possible even when the auxiliary information dimension is much less than the dimension of the true latent representation.

**Keywords:** disentanglement, non-linear independent component analysis

## 1. Introduction

Representation learning (Bengio et al., 2013) aims to extract low dimensional representations from high dimensional complex datasets. The hope is that if these representations succinctly capture factors of variation that describe the high dimensional data (e.g., extract features characterizing the shape of an object in an image), then these representations can be leveraged to achieve good performance on new downstream tasks with minimal supervision. Large scale pre-trained language models demonstrate the major success of representation learning based approaches (Brown et al., 2020; Wei et al., 2021; Radford et al., 2021). However, we should look at these results with a dose of caution, as neural networks have also been shown to fail often at out-of-distribution generalization (Beery et al., 2018; Geirhos et al., 2020; Peyrard et al., 2021). To address out-of-distribution

---

[*] Equal Contribution

generalization failures, recent works (Schölkopf, 2019; Schölkopf et al., 2021; Wang and Jordan, 2021) have argued in favour of incorporating causal principles into standard training paradigms—supervised (Arjovsky et al., 2019) and unsupervised (von Kügelgen et al., 2021). The issue is that the current deep learning paradigm does not imbibe and exploit key principles of causality (Pearl, 2009; Schölkopf, 2019)—invariance principle, independent causal mechanisms principle, and causal factorization. This is because the traditional causal inference requires access to structured random variables whose distributions can be decomposed using causal factorization, which is impossible with complex datasets such as images or text. Therefore, to leverage the power of deep learning and causal principles, we first need to disentangle raw datasets to obtain the causal representations that generated the data, and then exploit tools from causal structure learning to pin down the relationships between the representations. (Ke et al., 2019; Brouillard et al., 2020).

It has been shown that the general process of disentanglement is impossible in the absence of side knowledge of the structure of the data generation process (Hyvärinen and Pajunen, 1999; Locatello et al., 2019). However, under additional structural assumptions on the data generation process, it is possible to invert the data generation process and recover the underlying factors of variation (Hyvarinen and Morioka, 2016). Recently, there have been works (Hyvarinen et al., 2019; Khemakhem et al., 2020a) which present a general framework that relies on auxiliary information (e.g., labels, timestamps) to disentangle the latents. While existing works (Hyvarinen et al., 2019; Khemakhem et al., 2020a) have made remarkable progress in the field of disentanglement, these works make certain key assumptions highlighted below that we significantly depart from.

- **Labels cause the latent variables.** In supervised learning datasets, there are two ways to think about the data generation process—a) labels cause the latent variables and b) latent variables cause the labels. (Schölkopf et al., 2012) argue for the former view, i.e., labels generate the latents, while (Arjovsky et al., 2019) argue for the latter view, i.e., latents generate the label (see Figure 1). Current non-linear ICA literature (Khemakhem et al., 2020a; Hyvarinen et al., 2019) assumes the label knowledge renders latent factors of variation conditionally independent, hence it is compatible with the former perspective (Schölkopf et al., 2012). But the latter view might be more natural for the setting where a human assigns labels based on the underlying latent factors. Our goal is to enable disentanglement for this case when the latent variables cause the labels (Arjovsky et al., 2019).

- **Amount of auxiliary information.** Existing works (Khemakhem et al., 2020a) (Theorem 1), Khemakhem et al. (2020b) require a lot of auxiliary information, e.g., the number of label classes should be twice the total dimension of the latent factors of variation to guarantee disentanglement. We seek to enable disentanglement with lesser auxiliary information.

**Contributions.** We consider the following data generation process – latent factors generate the observations (raw features) and the labels for multiple tasks, where the latent factors are mutually independent. We study a natural extension of the standard empirical risk minimization (ERM) (Vapnik (1992)) paradigm. The most natural heuristic for learning representations is to train a neural network using ERM and use the output from the representation layer before the final layer. In this work, we propose to add a constraint on ERM to facilitate disentanglement – all the components of the representation layer must be mutually independent. Our main findings for the representations learned by the constrained ERM are summarized below.

- If the number of tasks is at least equal to the dimension of the latent variables, and the latent variables are not Gaussian, then we can recover the latent variables up to permutation and scaling.

• If we only have a single task and the latent variables come from an exponential family whose log-density can be expressed as a polynomial, then under further constraints on both the learner's inductive bias and the function being inverted, we can recover the latent variables up to permutation and scaling.

• To implement constrained ERM, we propose a simple two-step approximation. In the first step, we train a standard ERM based model, and in the subsequent step we carry out linear ICA (Comon, 1994) on the representation extracted from ERM. We carry out experiments with the above procedure for regression and classification. Our experiments show that even with the approximate procedure, it is possible to recover the true latent variables up to permutation and scaling when the number of tasks is smaller than the latent dimension.

## 2. Related work

**Non-linear ICA with auxiliary information.** We first describe the works in non-linear ICA where the time index itself serves as auxiliary information. Hyvarinen and Morioka (2016) showed that if each component of the latent vector evolves independently and follows a non-stationary time series without temporal dependence, then identification is possible for non-linear ICA. In contrast, Hyvarinen and Morioka (2017) showed that if the latent variables are mutually independent, with each component evolving in time following a stationary time series with temporal dependence, then also identification is possible. (Khemakhem et al., 2020a; Hyvarinen et al., 2019; Khemakhem et al., 2020b) further generalized the previous results. In these works, instead of using time, the authors require observation of auxiliary information. Note that (Hyvarinen and Morioka, 2017; Hyvarinen et al., 2019; Khemakhem et al., 2020a) have a limitation that the auxiliary information renders latent variables conditionally independent. This assumption was relaxed to some extent in (Khemakhem et al., 2020b), however, the model in (Khemakhem et al., 2020b) is not compatible with the data generation perspective that we consider, i.e., latent variables cause the labels. Recently, (Roeder et al., 2020) studied representation identification for classification and self-supervised learning models, where it was shown that if there are sufficient number of class labels (at least as many as the dimension of the latent variables), then the representations learned by neural networks with different initialization are related by a linear transformation. Their work does not focus on recovering the true latent variables and instead studies whether neural networks learn similar representations across different seeds.

**Other works.** In another line of work (Locatello et al., 2020; Shu et al., 2019), the authors study the role of weak supervision in assisting disentanglement. Note this line of work is different from us, as these models do not consider labelled supervised learning datasets, bur rather use different sources of supervision. For example, in (Locatello et al., 2020) the authors use multiple views of the same object as a form of weak supervision.

## 3. Problem Setup

### 3.1. Data generation process

Before we describe the data generation process we use, we give an example of the data generation process that is compatible with the assumptions in (Khemakhem et al., 2020a).

$$Y \leftarrow \mathsf{Bernoulli}\left(\frac{1}{2}\right) \qquad Z \leftarrow \mathcal{N}(Y\mathbf{1}, \mathsf{I}) \qquad X \leftarrow g(Z) \tag{1}$$

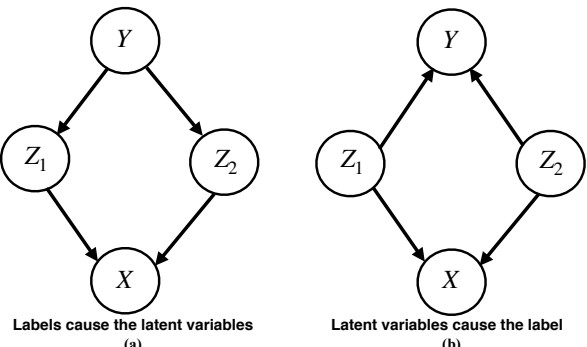

Figure 1: (a) Data generation process in Khemakhem et al. (2020b); (b) Data generation process studied in this work.

where Bernoulli($\frac{1}{2}$) is the uniform Bernoulli distribution over $\{0, 1\}$, $\mathcal{N}$ is normal distribution, $\mathbf{1} \in \mathbb{R}^d$ is the vector that together with the label $Y$ selects the mean of the latent $Z$, $\mathsf{I}$ is a $d$ dimensional identity matrix, $g$ is a bijection that generates the observations $X$. In Figure 1 (a), we show the causal directed acyclic graph (DAG) corresponding to the above data generation process, where the labels cause the latent variables (Schölkopf et al., 2012). Most of the current non-linear ICA models are only compatible with this view of the data generation process. This may be valid for some settings, but it is not perfectly suited for human labelled datasets where a label is assigned based on the underlying latent factors of variations. Hence, we focus on the opposite perspective (Arjovsky et al., 2019) when the latent variables generate the labels. The assumptions regarding the data generation process studied in our work are defined formally ahead.

**Assumption 1** *The data generation process for regression is described as*

$$
\begin{aligned}
Z &\leftarrow h(N_Z) \\
X &\leftarrow g(Z) \\
Y &\leftarrow \Gamma Z + N_Y
\end{aligned}
\tag{2}
$$

*where $N_Z \in \mathbb{R}^d$ is noise, $h : \mathbb{R}^d \to \mathbb{R}^d$ generates $Z \in \mathbb{R}^d$ (the components of latent variable $Z$ are mutually independent, non-Gaussian, and have finite second moments), $g : \mathbb{R}^d \to \mathbb{R}^d$ is a bijection that generates the observations $X$, $\Gamma \in \mathbb{R}^{k \times d}$ is a matrix that generates the label $Y \in \mathbb{R}^k$ and $N_Y \in \mathbb{R}^k$ is the noise vector ($N_Y$ is independent of $Z$ and $\mathbb{E}[N_Y] = 0$).*

*Note that $d$ is the dimension of the latent representation, and $k$ corresponds to the number of tasks. We adapt the data generation process to multi-task classification as follows by changing the label generation process.*

$$
Y \leftarrow \mathsf{Bernoulli}\Big(\sigma\big(\Gamma Z\big)\Big),
\tag{3}
$$

*where ($\sigma$) corresponds to the sigmoid function applied elementwise to $\Gamma Z$ and outputs the probabilities of the tasks, and $\mathsf{Bernoulli}$ operates on these probabilities elementwise to generate the label vector $Y \in \{0, 1\}^k$ for the $k$ tasks.*

We also contrast the DAGs of the two data generation processes in Figure 1. The nature of the auxiliary information (labels) in the data generation process we study (Assumption 1) is very different from the one in prior works (equation 1); conditioning on the auxiliary information (labels) in our data generation process does not make the latent variables independent. Our setting could be interpreted as multi-task supervised learning, where the downstream task labels serve as the auxiliary information generated from shared latent variables.

**Remark.** The classic non-identifiability results in Locatello et al. (2019) and Hyvärinen and Pajunen (1999) assumed that the latent factors of variation are all mutually independent. These results implied that without some knowledge, e.g., auxiliary information (labels), it is impossible to disentangle the latent variables. While (Khemakhem et al., 2020a) showed that it is possible to succeed in the presence of auxiliary information, their data generation process assumes that the latent variables are not mutually independent and thus is not consistent with assumptions in (Locatello et al., 2019) and (Hyvärinen and Pajunen, 1999). Whereas our work shows that the auxiliary information helps in the case considered in (Locatello et al., 2019) and (Hyvärinen and Pajunen, 1999), since the auxiliary information is introduced downstream of the latent variables.

### 3.2. Identifiability

Our objective is to learn the model $g^{-1}$ (or the generator $g$) from the observed data and labels pairs $(X, Y)$, such that for new observations $X$ we can recover the latent variables $Z$ that generated $X$. We can not always learn the exact latent variables $Z$ but may only identify them to some degree. Let us denote the model learned as $\tilde{g}^{-1}$ (with its inverse $\tilde{g}$). We now describe a general notion (commonly used in the literature) of identification for the learned map $\tilde{g}^{-1}$ with respect to the true map $g^{-1}$.

**Definition 1** *Identifiability up to $\mathcal{A}$. If the learned map $\tilde{g}^{-1}$ and the true map $g^{-1}$ are related by some bijection $a \in \mathcal{A}$, such that $\tilde{g}^{-1} = a \circ g^{-1}$ (or equivalently $\tilde{g} = g \circ a^{-1}$), then $\tilde{g}^{-1}$ is said to learn $g^{-1}$ up to bijections in $\mathcal{A}$. We denote this $\tilde{g}^{-1} \sim_{\mathcal{A}} g^{-1}$.*

In the above definition, if $\mathcal{A}$ was the set of all the permutation maps $\mathcal{P}$, then $\tilde{g}^{-1} \sim_{\mathcal{P}} g^{-1}$ denotes identification up to permutations. Permutation identification guarantees that we learn the true latent vector but do not learn the indices of the true latent, which is not important for many downstream tasks. In other words, identification up to permutations of the latent variables is the gold standard for identification. Our aim is to identify the latent variables $Z$ by inverting the data generating process (learning $g^{-1}$) up to permutations.

## 4. Identification via independence constrained ERM

The previous section established our objective of learning the model $g^{-1}$ (or the generator $g$) from the observed data $(X, Y)$. We first train a supervised learning model to predict the labels $Y$ from $X$. For the rest of the work, we will assume that the predictor we learn takes the form $\Theta \circ \Phi$, where $\Theta \in \mathbb{R}^{k \times d}$ is a linear predictor that operates on the representation $\Phi : \mathbb{R}^d \to \mathbb{R}^d$. As a result, the hypothesis space of the functions that the learner searches over has two parts: $\Theta \in \mathcal{H}_\Theta$ corresponding to the hypothesis class of linear maps, and $\Phi \in \mathcal{H}_\Phi$, where $\mathcal{H}_\Phi$ corresponds to the hypothesis class over the representations. We measure the performance of the predictor on an instance $(X, Y)$ using the loss $\ell\big(Y, \Theta \circ \Phi(X)\big)$ (mean square error for regression, cross-entropy loss for classification).

We define the risk achieved by a predictor $\Theta \circ \Phi$ as $R(\Theta \circ \Phi) = \mathbb{E}\Big[\ell\big(Y, \Theta \circ \Phi(X)\big)\Big]$, where the expectation is taken with respect to the data $(X, Y)$.

**Independence Constrained ERM (IC-ERM):** The representations ($\Theta$) learnt by ERM as described above have no guarantee of recovering the true latent variables up to permutations. Hence, we propose a new objective where we want the learner to carry out constrained empirical risk minimization, where the constraint is placed on the representation layer that all components of the representation are mutually independent. We provide the formal definition of mutual independence for the convenience of the reader below.

**Definition 2** *Mutual independence. A random vector $V = [V_1, \cdots, V_d]$ is said to be mutually independent if for each subset $\mathcal{M} \subseteq \{1, \cdots, d\}$ we have $P(\{V_i\}_{i \in \mathcal{M}}) = \prod_{i \in \mathcal{M}} P(V_i)$.*

We state the proposed independence constrained ERM (IC-ERM) objective formally as follows:

$$\min_{\Theta \in \mathcal{H}_\Theta, \Phi \in \mathcal{H}_\Phi} R(\Theta \circ \Phi) \text{ s.t. } \Phi(X) \text{ is mutually independent (Definition 2)} \tag{4}$$

We now state theorems that show the IC-ERM learning objective would recover the true latent variables up to permutations under certain assumptions. It is intuitive that more auxiliary information/numbers of tasks ($k$) should help us to identify the latent variables as they are shared across these different tasks. Hence, we first state identification guarantees for IC-ERM when we have sufficient tasks, and then discuss the difficult cases when we have few tasks.

### 4.1. Identification when number of tasks is equal to the latent data dimension

We first study the setting when the number of tasks $k$ is equal to the dimension of the latent variables $d$. Before we describe the theorem, we state assumptions on the hypothesis class of the representations ($\mathcal{H}_\Phi$) and the classifier ($\mathcal{H}_\Theta$).

**Assumption 2** *Assumption on $\mathcal{H}_\Phi$ and $\mathcal{H}_\Theta$. For the true solutions ($g^{-1}$, $\Gamma$), we have $g^{-1} \in \mathcal{H}_\Phi$ and $\Gamma \in \mathcal{H}_\Theta$. For the case when $k = d$, the set $\mathcal{H}_\Theta$ corresponds to the set of all invertible matrices.*

The above assumption ensures that the true solutions $g^{-1}$ and $\Gamma$ are in the respective hypothesis classes that the learner searches over. Also, the invertibility assumption on the hypothesis in $\mathcal{H}_\Theta$ is needed to ensure that we do not have redundant tasks for the identification of the latent variables. Under the above assumption and the assumptions of our data generation process (1), we state the following identification result for the case when k=d.

**Theorem 1** *If Assumptions 1, 2 hold and the number of tasks $k$ is equal to the dimension of the latent $d$, then the solution $\Theta^\dagger \circ \Phi^\dagger$ to IC-ERM (4) with $\ell$ as square loss for regression and cross-entropy loss for classification identifies true $Z$ up to permutation and scaling.*

The proof for the same is available in Appendix Section A. This implies that for the DAGs in Figure 1 (b), it is possible to recover the true latents up to permutation and scaling. This result extends the current disentanglement guarantees (Khemakhem et al., 2020b) that exist for models where labels cause the latent variables (latent variables are conditionally independent) to the settings where latent variables cause the label (latent variables are not conditionally independent). In multi-task learning literature (Caruana, 1997; Zhang and Yang, 2017), it has been argued that learning across multiple

tasks with shared layers leads to internal representations that transfer better. The above result states the conditions when the ideal data generating representation shared across tasks can be recovered.

**Remark.** Since we use a linear model for the label generation process, one can ask what happens if we apply noisy linear ICA techniques (Davies, 2004; Hyvarinen, 1999) on the label itself (when $k = d$) to recover $Z$ followed by a regression to predict $Z$ from $X$. Noisy linear ICA require the noise distribution to be Gaussian and would not work when $N_Y$ is not a Gaussian. Since we do not make such distributional assumptions on $N_Y$, we cannot rely on noisy linear ICA on labels.

## 4.2. Identification when the number of tasks is less than the dimensions of the latent

In this section, we study the setting when the number of tasks $k$ is equal to one. Since this setting is extreme, we need to make stronger assumptions to show latent identification guarantees. Before we lay down the assumptions, we provide some notation. Since we only have a single task, instead of using the matrix $\Gamma \in \mathbb{R}^{k \times d}$, we use $\gamma \in \mathbb{R}^d$ to signify the coefficients that generate the label in the single task setting. We assume each component of $\gamma$ is non-zero. In the single task setting for regression problems, the label generation is written as $Y \leftarrow \gamma^{\mathsf{T}} Z + N_Y$, and the rest of the notation is the same as the data generation process in Assumption 1. We rewrite the data generation process in Assumption 1 for the single task case in terms of normalized variables $U = Z \odot \gamma$.

**Assumption 3** *The data generation process for regressions is described as*

$$
\begin{aligned}
Z &\leftarrow h(N_Z) \\
Y &\leftarrow \mathbf{1}^{\mathsf{T}} U + N_Y, \\
X &\leftarrow g^{'}(U),
\end{aligned}
\tag{5}
$$

*where $g^{'}(U) = g(U \odot \frac{1}{\gamma})$, where $U \odot \frac{1}{\gamma} = [\frac{U_1}{\gamma_1}, \cdots , \frac{U_d}{\gamma_d}]$. We assume that all the components of $U$ are mutually independent and identically distributed (i.i.d.).*

Note that $g^{'}$ is invertible since $g$ is invertible and each element of $\gamma$ is also non-zero. Hence, for simplicity, we can deal with the identification of $U$. If we identify $U$ up to permutation and scaling, then $Z$ is automatically identified up to permutation and scaling. The predictor we learn is a composition of linear predictor $\theta$ and a representation $\Phi$, which is written as $\theta \circ \Phi(X) = \theta^{\mathsf{T}} \Phi(X)$. The learner searches for $\theta$ in the set $\mathcal{H}_\theta$, where $\mathcal{H}_\theta$ consists of linear predictors with all non-zero components, and $\Phi$ in the set $\mathcal{H}_\Phi$.

We can further simplify the predictor as follows: $\theta^{\mathsf{T}} \Phi(X) = \mathbf{1}^{\mathsf{T}}(\Phi(X) \odot \theta)$, where $\Phi(X) \odot \theta$ is component-wise multiplication expressed as $\Phi(X) \odot \theta = [\Phi_1(X) * \theta_1, \cdots , \Phi_d(X) * \theta_d]$. Therefore, instead of searching over $\mathcal{H}_\theta$ such that all components of $\theta$ are non-zero, we can fix $\mathcal{H}_\theta = \{\mathbf{1}\}$ and carry out the search over representations $\mathcal{H}_\Phi$ only. For the rest of the section, without loss of generality, we assume the predictor is of the form $\mathbf{1} \circ \Phi(X) = \mathbf{1}^{\mathsf{T}} \Phi(X)$. We restate the IC-ERM (4) with this parametrization and an additional constraint that all components are now required to be independent and identically distributed (i.i.d.). We provide a formal definition for the convenience of the reader below, where $\overset{d}{=}$ denotes identical in distribution.

**Definition 3** *Independent & Identically Distributed (i.i.d.). A random vector $V = [V_1, \cdots , V_d]$ is said to be i.i.d. if 1) $V_i(X) \overset{d}{=} V_j(X) \; \forall i, j \in \{1, \cdots , d\}$ 2) $P(\{V_i\}_{i \in \mathcal{M}}) = \prod_{i \in \mathcal{M}} P(V_i) \; \forall \mathcal{M} \subseteq \{1, \cdots , d\}$.*

The reparametrized IC-ERM (4) constraint is stated as follows.

$$\min_{\Phi \in \mathcal{H}_\Phi} R(\mathbf{1} \circ \Phi) \quad \text{s.t. } \Phi(X) \text{ is i.i.d. (Definition 3)} \tag{6}$$

Next, we state the assumptions on each component of $U$ (recall each component of $U$ is i.i.d. from Assumption 3) and $\mathcal{H}_\Phi$ under which we show that the reparametrized IC-ERM objective (equation (6)) recovers the true latent variables $U$ up to permutations. We assume each component of $U$ is a continuous random variable with probability density function (PDF) $r$. Define the support of each component of $U$ as $\mathcal{S} = \{u \mid r(u) > 0, u \in \mathbb{R}\}$. Define a ball of radius $\sqrt{2}p$ as $\mathcal{B}_p = \{u \mid |u|^2 \leq 2p^2, u \in \mathbb{R}\}$.

**Assumption 4** *Each component of $U$ is a continuous random variable from the exponential family with probability density $r$. $\log(r)$ is a polynomial with degree $p$ (where $p$ is odd) written as*

$$\log\big(r(u)\big) = \sum_{k=0}^{p} a_k u^k$$

*where the absolute value of the coefficients of the polynomial are bounded by $a_{\max}$, i.e., $|a_k| \leq a_{\max}$ for all $k \in \{1, \cdots, p\}$, and the absolute value of the coefficient of the highest degree term is at least $a_{\min}$, i.e., $|a_p| \geq a_{\min} > 0$. The support of $r$ is sufficiently large that it contains $\mathcal{B}_p$, i.e., $\mathcal{B}_p \subseteq \mathcal{S}$. Also, the moment generation function of each component $i$ of $U$, $M_{U_i}(t)$, exists for all $t$.*

**Remark on the PDFs under the above assumption.** The above assumption considers distributions in the exponential family, where the log-PDF can be expressed as a polynomial. Note that as long as the support of the distribution is bounded, every polynomial with bounded coefficients leads to a valid PDF (i.e., it integrates to one) and we only need to set the value of $a_0$ appropriately.

We now state our assumptions on the hypothesis class $\mathcal{H}_\Phi$ that the learner searches over. Observe that $\Phi(X)$ can be written as $h(U) = \Phi\big(g'(U)\big)$ (since $X = g'(U)$). We write the set of all the maps $h$ constructed from composition of $\Phi \in \mathcal{H}_\Phi$ and $g'$ as $\mathcal{H}_\Phi \circ g'$. Define $w(u_1, \cdots, u_d) = \log\Big(\big|\det\big[J(h(u_1, \cdots, u_d))\big]\big|\Big)$, where det is the determinant, $J(h(u_1, \cdots, u_d))$ is the Jacobian of $h$ computed at $(u_1, \cdots, u_d)$. The set of all the $w$'s obtained from all $h \in \mathcal{H}_\Phi \circ g'$ is denoted as $\mathcal{W}$

**Assumption 5** $\mathcal{H}_\Phi$ *consists of analytic bijections. For each $\Phi \in \mathcal{H}_\Phi$, the moment generating function of each component $i \in \{1, \cdots, d\}$, $\Phi_i(X)$, denoted as $M_{\Phi_i(X)}(t)$ exists for all $t$. Each $w \in \mathcal{W}$ is a finite degree polynomial with degree at most $q$, where the absolute values of the coefficients in the polynomial are bounded by $b_{\max}$.*

Define $p_{\min} = \max\left\{\kappa \log(8(d-1)), \frac{4a_{\max}(d-1)}{a_{\min}}, \frac{\log\left(\frac{4b_{\max} * n_{\text{poly}}}{a_{\min}}\right)}{2} + q\right\}$, where $n_{\text{poly}}$ is the maximum number of non-zero coefficients in any polynomial $w \in \mathcal{W}$, $\kappa$ is small constant (see Appendix C).

**Theorem 2** *If the Assumptions 3, 4, 5 hold, $(g')^{-1} \in \mathcal{H}_\Phi$, and $p$ is sufficiently large ($p \geq p_{\min}$), then the solution $\Phi^\dagger(X)$ of reparametrized IC-ERM objective (6) recovers the true latent $U$ in the data generation process in Asssumption 3 up to permutations.*

**Proof sketch.** The complete proof is available in Appendix Section C and we provide an overview here. We use the optimality condition that the prediction made by the learned model, $\mathbf{1}^{\mathsf{T}}h(U)$, exactly matches the true mean, $\mathbf{1}^{\mathsf{T}}U$, along with the constraints that each component of $U$ are i.i.d. and each component of $h(U)$ are i.i.d., to derive that the distributions of $U$ and $h(U)$ are the same. We substitute this condition in the change of variables formula that relates the densities of $U$ and $h(U)$. Using the Assumption 5, we can show that if the highest absolute value of $U$ and the highest absolute value of $h(U)$ are not equal, then the term with the highest absolute value among $U$ and $h(U)$ will dominate, leading to contradiction in the identity obtained by change of variables formula. Based on this, we can conclude that the highest absolute value of $U$ and the highest absolute value of $h(U)$ must be equal. We iteratively apply this argument to show that all the absolute values of $U$ and $h(U)$ are related by permutation. We can extend this argument to the actual values instead of absolute values. Since $h$ is analytic we argue that the relationship of permutation holds in a neighborhood. Then we use properties of analytic functions (Mityagin, 2015) to conclude that the relationship holds on the entire space.

**Why does the bound on $p$ grow linearly in $d$?** We provide some geometric intuition into why the degree of the polynomial ($p$) of the log-PDF ($\log(r)$) needs to be large. If the dimension of the latent space $d$ is large, then the second term in $p_{\mathsf{min}}$ dominates, i.e., $p$ has to grow linearly in $d$. The simplification in the proof yields that the mapping $h$ must satisfy the following condition – $\|u\|_k = \|h(u)\|_k$ for all $k \in \{1, \cdots, p\}$, where $\|\cdot\|_k$ is the $k^{th}$ norm. Hence, $h$ is a bijection that preserves all norms up to the $p^{th}$ norm. If $U$ is 2 dimensional, then the a bijection $h$ that preserves the $\ell_1$ norm and the $\ell_2$ norm is composed of permutations and sign-flips. In general, since $U$ is $d$ dimensional, we need at least $d$ constraints on $h$ in the form $\|u\|_k = \|h(u)\|_k$ and thus $p \geq d$, which ensure that the only map that satisfies these constraints is composed of permutations and sign-flips.

**Significance and remarks on Theorem 2.** Theorem 2 shows that if we use IC-ERM principle, i.e., constraint the representations to be independent, then we continue to recover the latents even if the number of tasks is small. We can show that the above theorem also extends to binary classification. We admit that strong assumptions were made to arrive at the above result, while other assumptions such as bound on $p$ growing linearly in $d$ seem necessary, but we would like to remind the reader that we are operating in the extreme single task regime. In the previous section, when the number of tasks was equal to the dimension of the latent (when we have sufficiently many tasks), we had shown the success of the IC-ERM (4) objective (Theorem 1) for identification of latent variables with much fewer assumptions. In contrast, in Theorem 2, we saw that with more assumptions on the distribution we can guarantee latent recovery with even one task. If we are in the middle, i.e., when the number of tasks is between one and the dimension of the latent, then the above Theorem 2 says we only require the assumptions (Assumptions 3, 4, 5) to hold for at least one task.

**Note on the case k>d**: We did not discuss the case when the number of tasks is greater than the dimension of the latent variables. This is because we can select a subset $S'$ of tasks, such that $|S'| = d$ and then proceed in a similar fashion as Theorem 1. This question arises commonly in linear ICA literature, and selecting a subset of tasks is the standard practice.

## 5. ERM-ICA: Proposed implementation for independence constrained ERM

In the previous section, we showed the identification guarantees with the IC-ERM objective. However, solving this objective is non-trivial, since we need to enforce independence on the representa-

tions learnt. We propose a simple two step procedure as an approximate approach to solve the above problem. The learner first carries out standard ERM stated as

$$\Theta^{\dagger}, \Phi^{\dagger} \in \underset{\Theta \in \mathcal{H}_{\Theta}, \Phi \in \mathcal{H}_{\Phi}}{\arg\min} R(\Theta \circ \Phi) \tag{7}$$

The learner then searches for a linear transformation $\Omega$ that when applied to $\Phi^{\dagger}$ yields a new representation with mutually independent components. We state this as follows. Find an invertible $\Omega \in \mathbb{R}^{d \times d}$ such that

$$Z^{\dagger} = \Omega \circ \Phi^{\dagger}(X) \text{ where the components of } Z^{\dagger} \text{ are mutually independent} \tag{8}$$

Note that a solution to the above equation (8) does not always exist. However, if we can find a $\Omega$ that satisfies the above (8), then the classifier $\Theta \circ \Omega^{-1}$ and the representation $\Omega \circ \Phi^{\dagger}(X)$ together solve the IC-ERM (4) assuming $\Theta \circ \Omega^{-1} \in \mathcal{H}_{\Theta}$ and $\Omega \circ \Phi^{\dagger} \in \mathcal{H}_{\Phi}$. To find a solution to the equation (8) we resort to the approach of linear ICA (Comon, 1994). The approach has two steps. We first whiten $\Phi^{\dagger}(X)$. [1] Define $V$ to be the covariance matrix of $\Phi^{\dagger}(X)$. If the covariance $V$ is invertible, [2] then the eigendecomposition of $V$ is given as $V = U\Lambda^2 U^{\mathsf{T}}$. We obtain the whitened data $\Phi^{*}(X)$ as follows $\Phi^{*}(X) = \Lambda^{-1}U^{\mathsf{T}}\Phi^{\dagger}(X)$. Consider a linear transformation of the whitened data and denote it as $Z^{*} = \Omega \circ \Phi^{*}(X)$ and construct another vector $Z^{'}$ such that its individual components are all independent and equal in distribution to the corresponding components in $Z^{*}$. Our goal is to find an $\Omega$ such that the mutual information between $Z^{*}$ and $Z^{'}$ is minimized. To make dependence on $\Omega$ explicit, we denote the mutual information between $Z^{'}$ and $Z^{*}$ as $I(\Omega \circ \Phi^{*}(X))$. We state this as the following optimization

$$\Omega^{\dagger} \in \underset{\Omega, \Omega \text{ is invertible}}{\arg\min} I(\Omega \circ \Phi^{*}(X)) \tag{9}$$

We denote the above two step approximation method as ERM-ICA and summarize it below:

- **ERM Phase:** Learn $\Theta^{\dagger}, \Phi^{\dagger}$ by solving the ERM objective (Eq: 7).

- **ICA Phase:** Learn $\Omega^{\dagger}$ by linear ICA (Eq: 9) on the representation from ERM Phase ($\Phi^{\dagger}$).

The above ERM-ICA procedure outputs a classifier $\Theta^{\dagger} \circ (\Omega^{\dagger})^{-1}$ and representation $\Omega^{\dagger} \circ \Phi^{\dagger}$ that is an approximate solution to the IC-ERM problem (4). While the proposed ERM-ICA procedure is a simple approximation, we do not know of other works that have investigated this approach theoretically and experimentally for recovering the latents. Despite its simplicity, we can prove (similar to Theorem 1) that when the number of tasks is equal to the dimension of the latent variable, ERM-ICA can recover the latent variables up to permutation and scaling.

**Theorem 3** *If Assumptions 1, 2 hold and the number of tasks $k$ is equal to the dimension of the latent $d$, then the solution $\Omega^{\dagger} \circ \Phi^{\dagger}$ to ERM-ICA ((7), (9)) with $\ell$ as square loss for regression and cross-entropy loss for classification identifies true $Z$ up to permutation and scaling.*

---

1. For simplicity, we assume $\Phi^{\dagger}(X)$ is zero mean, although our analysis extends to the non-zero mean case as well. We also assume $\Phi^{\dagger}(X)$ has finite second moments.
2. If it is not invertible, then we first need to project $\Phi^{\dagger}(X)$ into the range space of $V$

The proof of the above theorem can be found in Appendix Section B. We leave the theoretical analysis of the ERM-ICA approach for the single task case for future work, but we do show its performance empirically for such scenarios in the evaluation section ahead. We believe that building better approximations to directly solve the IC-ERM, which do not involve a two-step approach like ERM-ICA approach is a fruitful future work.

## 6. Evaluation

### 6.1. Experiment Setup

#### 6.1.1. DATA GENERATION PROCESS

**Regression.** We use the data generation process described in Assumption 1. The components of $Z$ are i.i.d. and follow discrete uniform $\{0, 1\}$ distribution. Each element of the task coefficient matrix $\Gamma$ is i.i.d. and follows a standard normal distribution. The noise in the label generation is also standard normal. We use a 2-layer invertible MLP to model $g$ and follow the construction used in Zimmermann et al. (2021).[3] We carry out comparisons for three settings, $d = \{16, 24, 50\}$, and vary tasks from $k = \{\frac{d}{2}, \frac{3d}{4}, d\}$. The dataset size used for training and test is 5000 data points, along with a validation set of 1250 data points for hyper parameter tuning.

**Classification.** We use the data generation process described in Assumption 1. We use the same parameters and dataset splits as regression, except the labels are binary and sampled as follows: $Y \leftarrow \text{Bernoulli}(\sigma(\Gamma Z))$. Also, the noise in the $\Gamma$ sampling is set to a higher value (10 times that of a standard normal), as otherwise the Bayes optimal accuracy is much smaller.

#### 6.1.2. METHODS, ARCHITECTURE, AND OTHER HYPERPARAMETERS

We compare our method against two natural baselines. **a) ERM.** In this case, we carry out standard ERM (7) and use the representation learned at the layer before the output layer. **b) ERM-PCA.** In this case, we carry out standard ERM (7) and extract the representation learned at the layer before the output layer. We then take the extracted representation and transform it using principal component analysis (PCA). In other words, we analyze the representation in the eigenbasis of its covariance matrix. **c) ERM-ICA.** This is the main approach ((7), (8)) that approximates the IC-ERM objective (4). Here we take the representations learnt using ERM (7) and transform them using linear independent component analysis (ICA) (9).

We define mean square error and cross-entropy as our loss objectives for the regression and classification task respectively. In both settings, we use a two layer fully connected neural network and train the model using stochastic gradient descent. The architecture and the hyperparameter details for the different settings are provided in Appendix D. Also, the code repository can be accessed from the link [4] in the footnote.

#### 6.1.3. METRICS

The models are evaluated on the test dataset using the following two metrics.

- **Label ($Y$) prediction performance.** We use the average $R^2$ (coefficient of determination) and the average accuracy across tasks in the regression and classification task respectively. For

---

3. https://github.com/brendel-group/cl-ica/blob/master/main_mlp.py
4. Our Code Repository: https://github.com/divyat09/ood_identification

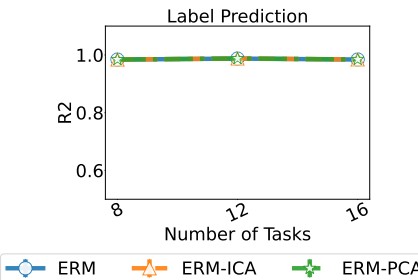 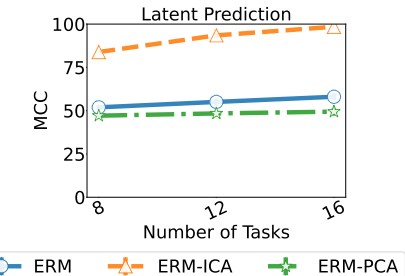

Figure 2: Comparison of label and latent prediction performance (regression, $d = 16$).

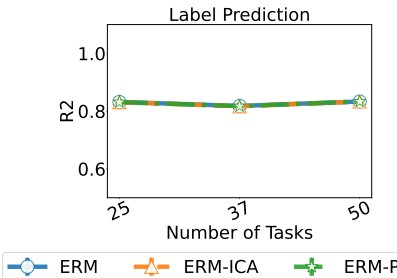 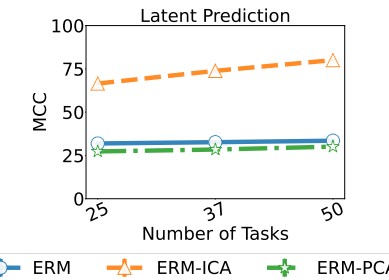

Figure 3: Comparison of label and latent prediction performance (regression, $d = 50$).

the ERM-PCA and ERM-ICA, we take final representations learnt by these methods and train a linear/logistic regression model to predict the label $Y$ for the regression/classification task. We check this metric to ensure that the representations do not lose any information about the label.

• **Latent ($Z$) prediction performance.** We use mean correlation coefficient (MCC), a standard metric used to measure permutation and scaling based identification (refer (Hyvarinen and Morioka, 2017; Zimmermann et al., 2021) for further details). The metric is computed by first obtaining the correlation matrix ($\rho(Z, \hat{Z})$) between the recovered latents $\hat{Z}$ and the true latents $Z$. Let's define $|\rho(Z, \hat{Z})|$ as the absolute values of the correlation matrix. Then we find a matching (assign each row to a column in $|\rho(Z, \hat{Z})|$) such that the average absolute correlation is maximized and return the optimal average absolute correlation. Intuitively, we find the optimal way to match the components of the predicted latent representation ($\hat{Z}$) and components of the true representation ($Z$). Notice that a perfect absolute correlation of one for each matched pair of components would imply identification up to permutations.

### 6.2. Results

**Regression.** Figure 2 and 3 show a comparison of the performance of the three approaches across $d = 16$ and $d = 50$ for various tasks. The results for the case of $d = 24$ are in the Appendix E.1 (due to space limits). In both cases, we find that the method ERM-ICA is significantly better than the other approaches in terms of guaranteeing permutation and scaling based identification. All three approaches have similar label prediction performance. We observe a similar trend for the case of $d = 24$ shown in the Appendix E.1.

**Classification.** Figure 4 and 5 show a comparison of the performance of the three approaches across $d = 16$ and $d = 24$ for various tasks. In both cases, we find that the method ERM-ICA is better than

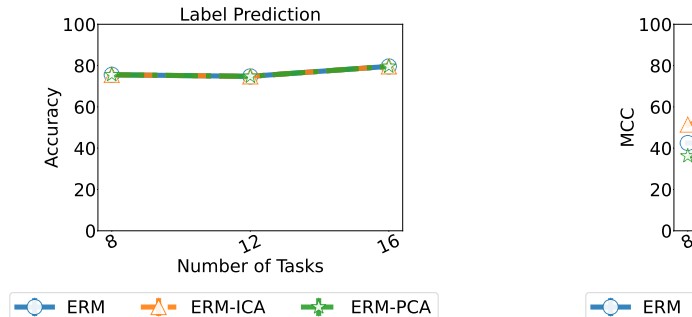

Figure 4: Comparison of label and latent prediction performance (classification, $d = 16$)

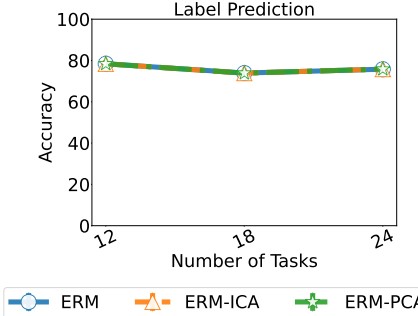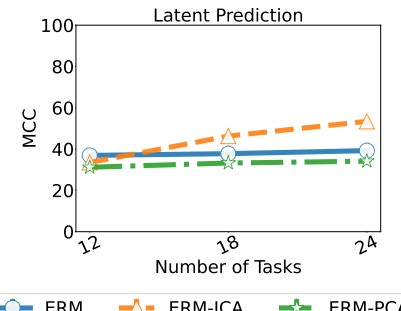

Figure 5: Comparison of label and latent prediction performance (classification, $d = 24$)

the other approaches in terms of guaranteeing permutation and scaling based identification, except in the case of 24 data dimensions with a total of 12 tasks. All three approaches have similar label prediction performance. For classification, unlike regression, the improvements are smaller and we do not see improvement for $d = 50$ (comparisons for the case of $d = 50$ are in the Appendix E.2). We see a worse performance in the classification setting because the ERM model does not learn the Bayes optimal predictor unlike regression.

**Discussion.** We have shown that ERM-ICA achieves significant improvement in latent recovery with much fewer tasks (up to $\frac{d}{2}$). However, in our theory we proved that under certain assumptions solving the reparametrized IC-ERM objective (Eq (6)) can achieve identification even with a single task. Note that we only approximate equation (6) with ERM-ICA, and if we build better approximations of the ideal approach (IC-ERM), then we can witness gains with even fewer tasks. We believe that building such approximations is a fruitful future work.

## 7. Conclusion

In this work, we analyzed the problem of disentanglement in a natural setting, where latent factors cause the labels, a setting not well studied in the ICA literature. We show that if ERM is constrained to learn independent representations, then we can have latent recovery from learnt representations even when the number of tasks is small. We propose a simple two step approximate procedure (ERM-ICA) to solve the constrained ERM problem, and show that it is effective in a variety of experiments. Our analysis highlights the importance of learning independent representations and motivates the development of further approaches to achieve the same in practice.

## Acknowledgments

We thank Dimitris Koukoulopoulos for discussions regarding the proof for indentification under a single task. Kartik Ahuja acknowledges the support provided by IVADO postdoctoral fellowship funding program. Ioannis Mitliagkas acknowledges support from an NSERC Discovery grant (RGPIN-2019-06512), a Samsung grant, Canada CIFAR AI chair and MSR collaborative research grant.

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

## Appendix A. Proof of Theorem 1: IC-ERM for the case k=d

**Theorem 1** *If Assumptions 1, 2 hold and the number of tasks $k$ is equal to the dimension of the latent d, then the solution $\Theta^\dagger \circ \Phi^\dagger$ to IC-ERM (4) with $\ell$ as square loss for regression and cross-entropy loss for classification identifies true $Z$ up to permutation and scaling.*

**Proof** Consider we are in the regression setting for the data generation process in Assumption 1. Therefore, using $X \leftarrow g(Z)$ and $Y \leftarrow \Gamma Z + N$, the risk of a predictor $f$ can be written as follows:

$$
\begin{aligned}
R(f) &= \mathbb{E}\big[\|Y - f(X)\|^2\big] \\
&= \mathbb{E}\big[\|\Gamma Z + N - f \circ g(Z)\|^2\big] \\
&= \mathbb{E}\Big[\|\Gamma Z - f \circ g(Z)\|^2\Big] + \mathbb{E}\big[\|N\|^2\big] - 2 * \mathbb{E}[(\Gamma Z - f \circ g(Z))^\mathsf{T} N] \\
&= \mathbb{E}\Big[\|\Gamma Z - f \circ g(Z)\|^2\Big] + \mathbb{E}\big[\|N\|^2\big] \quad \text{(since } Z \perp N \text{ and } \mathbb{E}[N] = 0)
\end{aligned}
\tag{10}
$$

From the above it is clear that $R(f) \geq \mathbb{E}\big[\|N\|^2\big]$ for all functions $f : \mathbb{R}^d \to \mathbb{R}^d$. Since $g^{-1} \in \mathcal{H}_\Phi$, $\Gamma \in \mathcal{H}_\Theta$ and $g^{-1}(X)$ has all mutually independent components, $\Gamma \circ g^{-1}$ satisfies the constraints in IC-ERM (4) and also achieves the lowest error possible, i.e., $R(\Gamma \circ g^{-1}) = \mathbb{E}[\|N\|^2]$. Consider any solution to constrained ERM in (4). The solution must satisfy the following equality except over a set of measure zero.

$$
\begin{aligned}
\Theta^\dagger \circ \Phi^\dagger(X) &= \Gamma Z \\
\implies \Phi^\dagger(X) &= (\Theta^\dagger)^{-1}\Gamma Z
\end{aligned}
\tag{11}
$$

Let us call $\Phi^\dagger(X) = Z^\dagger$ and $A = (\Theta^\dagger)^{-1}\Gamma$. Hence, the above equality becomes $Z^\dagger = AZ$, where all the components of $Z^\dagger$ are independent (Eq: (4)) and all the components of $Z$ are independent (Assumption 1). We will now argue that the matrix $A$ can be written as a permutation matrix times a scaling matrix. We first show that in each column of $A$ there is exactly one non-zero element. Consider column $k$ of $A$ denoted as $[A]_k$. Since $A$ is invertible all elements of the column cannot be zero. Now suppose at least two elements $i$ and $j$ of $[A]_k$ are non-zero. Consider the corresponding components of $Z^\dagger$. Since $Z_i^\dagger$ and $Z_j^\dagger$ are both independent and since $[A]_{ik}$ and $[A]_{jk}$ are both non-zero, from Darmois' theorem (Darmois, 1953) it follows that $Z_k$ is a Gaussian random variable. However, this leads to a contradiction as we assumed none of the random variables in $Z$ follow a Gaussian distribution. Therefore, exactly one element in $[A]_k$ is non-zero. We can say this about all the columns of $A$. No two columns will have the same row with a non-zero entry or otherwise $A$ would not be invertible. Therefore, $A$ can be expressed as a matrix permutation times a scaling matrix, where the scaling takes care of the exact non-zero value in the row and the permutation matrix takes care of the address of the element which is non-zero. This completes the proof. ∎

## Appendix B. Proof of Theorem 3: ERM-ICA for the case k=d

**Theorem 3** *If Assumptions 1, 2 hold and the number of tasks $k$ is equal to the dimension of the latent $d$, then the solution $\Omega^{\dagger} \circ \Phi^{\dagger}$ to ERM-ICA ((7), (9)) with $\ell$ as square loss for regression and cross-entropy loss for classification identifies true $Z$ up to permutation and scaling.*

**Proof** Although the initial half of the proof is identical to the proof of Theorem 1 we repeat it for clarity. Consider we are in the regression setting for the data generation process in Assumption 1. Therefore, using $X \leftarrow g(Z)$ and $Y \leftarrow \Gamma Z + N$, the risk of a predictor $f$ can be written as follows:

$$
\begin{aligned}
R(f) &= \mathbb{E}\big[\|Y - f(X)\|^2\big] \\
&= \mathbb{E}\big[\|\Gamma Z + N - f \circ g(Z)\|^2\big] \\
&= \mathbb{E}\Big[\|\Gamma Z - f \circ g(Z)\|^2\Big] + \mathbb{E}\big[\|N\|^2\big] - 2 * \mathbb{E}[(\Gamma Z - f \circ g(Z))^{\mathsf{T}} N] \\
&= \mathbb{E}\Big[\|\Gamma Z - f \circ g(Z)\|^2\Big] + \mathbb{E}\big[\|N\|^2\big] \quad \text{(since } Z \perp N \text{ and } \mathbb{E}[N] = 0)
\end{aligned}
\tag{12}
$$

From the above it is clear that $R(f) \geq \mathbb{E}\big[\|N\|^2\big]$ for all functions $f : \mathbb{R}^d \to \mathbb{R}^d$. Since $g^{-1} \in \mathcal{H}_{\Phi}$, $\Gamma \in \mathcal{H}_{\Theta}$ and $g^{-1}(X)$ has all mutually independent components, $\Gamma \circ g^{-1}$ satisfies the constraints in IC-ERM (4) and also achieves the lowest error possible, i.e., $R(\Gamma \circ g^{-1}) = \mathbb{E}[\|N\|^2]$.

Consider any solution to ERM in (7). The solution must satisfy the following equality except over a set of measure zero.

$$
\begin{aligned}
\Theta^{\dagger} \circ \Phi^{\dagger}(X) &= \Gamma Z \\
\implies \Phi^{\dagger}(X) &= (\Theta^{\dagger})^{-1} \Gamma Z
\end{aligned}
\tag{13}
$$

Since $\Phi^{\dagger}(X)$ is a linear combination of independent latents with at least one latent non-Gaussian (and also the latents have a finite second moment). We can use the result from (Comon, 1994) that states $\Omega^{\dagger}$ that solves equation (9) relates to $(\Theta^{\dagger})^{-1}$ as follows

$$
(\Omega^{\dagger})^{-1} = (\Gamma P \Lambda)^{-1} = \Lambda^{-1} P^{-1} \Gamma^{-1}
\tag{14}
$$

Substituting the above into (13) we get $\Phi^{\dagger}(X) = \Lambda^{-1} P^{-1} \Gamma^{-1} \Gamma Z = \Lambda^{-1} P^{-1} Z$. This completes the proof.

We can also carry out the same proof for the multi-task classification case. In multi-task classification we can write the condition for optimality as

$$
\sigma\Big(\Omega^{\dagger} \Phi^{\dagger}(X)\Big) = \sigma\Big(\Gamma Z\Big)
\tag{15}
$$

where sigmoid is applied separately to each element, and since the sigmoids are equal, this implies the individual elements are also equal. Therefore, $\Omega^{\dagger} \Phi^{\dagger}(X) = \Gamma Z$ and we can use the same analysis as the regression case from this point on. Also, note that we equated the sigmoids in first place, because the LHS corresponds to $\hat{P}(Y|X)$ and RHS corresponds to true $P(Y|X)$. ∎

### Appendix C. Proof of Theorem 2: IC-ERM for the case k=1

**Theorem 2** *If the Assumptions 3, 4, 5 hold, $(g')^{-1} \in \mathcal{H}_\Phi$, and $p$ is sufficiently large ($p \geq p_{\min}$), then the solution $\Phi^\dagger(X)$ of reparametrized IC-ERM objective (6) recovers the true latent $U$ in the data generation process in Asssumption 3 up to permutations.*

**Proof** We write $\Phi^\dagger(X) = [\Phi_1^\dagger(X), \cdots, \Phi_d^\dagger(X)]$. We call $\Phi_i^\dagger(X) = V_i$ and the vector $V = [V_1, \cdots, V_d] = [\Phi_1^\dagger(X), \cdots, \Phi_d^\dagger(X)]$.

Observe that since $(g')^{-1} \in \mathcal{H}_\Phi$, we can use the same argument used in the proof of Theorem 1 to conclude that $(g')^{-1}$ is a valid solution for the reparametrized IC-ERM objective (Eq: (6)). Notice that the terms $\Theta^\dagger$ and $\Gamma$ that appear in the proof of Theorem 1, they are equal to identity here due to the Assumption 3 and IC-ERM (6). Therefore, following the standard argument in Theorem 1, we can conclude that any solution $\Phi^\dagger(X)$ of reparametrized IC-ERM (6) satisfies the following:

$$\sum_i \Phi_i^\dagger(X) = \sum_i (g')_i^{-1}(X)$$
$$\sum_i V_i = \sum_i U_i \tag{16}$$

We write the moment generation function of a random variable $U_i$ as $M_{U_i}(t) = \mathbb{E}[e^{tU_i}]$. We substitute the moment generating functions to get the following identity.

$$\sum_i V_i = \sum_i U_i \implies \Pi_i M_{V_i}(t) = \Pi_i M_{U_i}(t) \tag{17}$$

Since $U_i \overset{d}{=} U_j$ and $V_i \overset{d}{=} V_j$, we can use $M_{U_i}(t) = M_{U_j}(t)$ and $M_{V_i}(t) = M_{V_j}(t)$ for all $t \in \mathbb{R}$ and simplify as follows:

$$\left(M_{V_i}(t)\right)^d = \left(M_{U_i}(t)\right)^d \implies M_{V_i}(t) = M_{U_i}(t)$$
$$\implies V_i \overset{d}{=} U_i, \forall i \in \{1, \cdots, d\} \tag{18}$$

In the second step of the above simplification, we use the fact that the moment generating function is positive. In the third step, we use the fact that if moment generating functions exist and are equal, then the random variables are equal in distribution (Feller, 2008). Having established that the distributions are equal, we now show that the random variables are equal up to permutations.

Since the vector $V$ is an invertible transform $h$ of $U$, where $V = h(U)$. We can write the pdf of $U$ in terms of $V$ as follows.

$$\prod_i p(u_i) = \prod_i p(v_i) \big| \det(\mathsf{J}(h(u))) \big|, \tag{19}$$

where $p$ corresponds to the pdf of $U_i$ (recall that pdfs of all components is the same). We take log on both sides of the above equation to get the following:

$$\sum_i \log(p(u_i)) = \sum_i \log(p(v_i)) + \log\left(\big| \det(\mathsf{J}(h(u))) \big|\right) \tag{20}$$

From Assumption 4, we substitute a polynomial for $\log(p(u)) = \sum_{k=0}^{p} a_k u^k$. From Assumption 5, we express this $\log\left(\left|\det(\mathsf{J}(h(u)))\right|\right) = \sum b_k \prod_i u_i^{\theta_k(i)}$. We substitute these polynomials into the above equation to get the following:

$$\sum_i \log(p(u_i)) - \sum_i \log(p(v_i)) - \log\left(\left|\det(\mathsf{J}(h))\right|\right) = 0$$

$$\implies \sum_{k=1}^{p} a_k \sum_{i=1}^{d} (u_i^k - v_i^k) - \sum_m b_m \prod_i u_i^{\theta_m(i)} = 0 \tag{21}$$

In the proof, we first focus on comparing the largest absolute value among $u$'s and largest absolute value among $v$'s. Without loss of generality, we assume that $|u_j| > |u_i|$ for all $i \neq j$ ($u_j$ is the largest absolute value among $u$'s) and $|v_r| > |v_i|$ for all $i \neq r$ ($v_r$ is the largest absolute value among $v$'s). Consider the setting when $|u_j| \geq p^2 > 1$ (these points exist in the support because of the Assumption 4). We can write $|u_j| = \alpha p^2$, where $\alpha \geq 1$.

There are three cases to further consider:

    a) $|v_r| > |u_j|$

    b) $|v_r| < |u_j|$

    c) $|v_r| = |u_j|$

We first start by analyzing the case b). Since all values of $|v_i|$ and $|u_i|$ are also strictly less than $|u_j|$, we have the following:

- $\exists\, c < 1$ such that $|u_i| < c\alpha p^2 \ \forall\, i \neq j$

- $|v_i| < c\alpha p^2 \ \forall\, i \in \{1, \cdots, d\}$, where $c < 1$.

We divide the identity in equation (21) by $u_j^p$ and separate the terms in such a way that only the term $a_p$ is in the LHS and the rest of the terms are pushed to the RHS. We show further simplification of the identity below.

$$|a_p| = \left| a_p \sum_{i=1}^{d-1} \left( \frac{u_i^p}{u_j^p} - \frac{v_i^p}{u_j^p} \right) + \sum_{k=1}^{p-1} a_k \sum_{i=1}^{d} \left( \frac{u_i^k}{u_j^p} - \frac{v_i^k}{u_j^p} \right) - \sum_m b_m \frac{1}{u_j^p} \prod_i u_i^{\theta_m(i)} \right|$$

$$\implies |a_p| \leq \left| a_p \sum_{i=1}^{d-1} \left( \frac{u_i^p}{u_j^p} - \frac{v_i^p}{u_j^p} \right) \right| + \left| \sum_{k=1}^{p-1} a_k \sum_{i=1}^{d} \left( \frac{u_i^k}{u_j^p} - \frac{v_i^k}{u_j^p} \right) \right| + \left| \sum_m b_m \frac{1}{u_j^p} \prod_i u_i^{\theta_m(i)} \right|$$

$$\implies 1 \leq \frac{1}{|a_p|} \left( \left| a_p \sum_{i=1}^{d-1} \left( \frac{u_i^p}{u_j^p} - \frac{v_i^p}{u_j^p} \right) \right| + \left| \sum_{k=1}^{p-1} a_k \sum_{i=1}^{d} \left( \frac{u_i^k}{u_j^p} - \frac{v_i^k}{u_j^p} \right) \right| + \left| \sum_m b_m \frac{1}{u_j^p} \prod_i u_i^{\theta_m(i)} \right| \right) \tag{22}$$

We analyze each of the terms in the RHS separately. The simplification of the first term yields the following expression.

$$\frac{1}{|a_p|} \left| a_p \sum_{i=1}^{d-1} \left( \frac{u_i^p}{u_j^p} - \frac{v_i^p}{u_j^p} \right) \right| \leq \frac{1}{|a_p|} |a_p| \sum_{i=1}^{d-1} \left( \left| \frac{u_i^p}{u_j^p} \right| + \left| \frac{v_i^p}{u_j^p} \right| \right)$$

$$\leq 2c^p(d-1) \tag{23}$$

The simplification of the second term in the RHS of the last equation in (22) yields the following expression.

$$
\begin{aligned}
\frac{1}{|a_p|}\Big| \sum_{k=1}^{p-1} a_k \sum_{i=1}^{d} \Big(\frac{u_i^k}{u_j^p} - \frac{v_i^k}{u_j^p}\Big)\Big| &\leq \frac{1}{|a_p|} \sum_{k=1}^{p-1} |a_k| \sum_{i=1}^{d} \Big(\Big|\frac{u_i^k}{u_j^p}\Big| + \Big|\frac{v_i^k}{u_j^p}\Big|\Big)\Big| \\
&\leq \sum_{k=1}^{p-1} 2|a_k| \sum_{i=1}^{d} \Big(\Big|\frac{u_j^k}{u_j^p}\Big|\Big)\Big| \\
&\leq \frac{1}{|a_p|}\Big(\sum_{k=1}^{p-1} 2|a_k|(d-1)\Big)\frac{1}{\alpha p^2} \\
&\leq \frac{2a_{\mathsf{max}}(d-1)}{a_{\mathsf{min}}p}
\end{aligned}
\tag{24}
$$

The simplification of the third term in the RHS of the last equation in (22) yields the following expression.

$$
\begin{aligned}
\frac{1}{|a_p|}\Big| \sum_m b_m \frac{1}{u_j^p} \prod_i u_i^{\theta_m(i)}\Big| &\leq \sum_m |b_m| \frac{1}{|u_j|^{(p-q)}} \\
&\leq \frac{b_{\mathsf{max}}}{a_{\mathsf{min}}} \frac{n_{\mathsf{poly}}}{(\alpha p^2)^{(p-q)}}
\end{aligned}
\tag{25}
$$

where $n_{\mathsf{poly}}$ corresponds to the number of non-zero terms in the polynomial expansion of the log-determinant. In the above simplification, we used the fact that $\sum_i \theta_m(i) \leq q$ and $|u_i| < |u_j|$. Analyzing the RHS in equations (23)-(25), we see that if $p$ becomes sufficiently large, the RHS becomes less than 1. This contradicts the relationship in equation (22). Therefore, all $|v_i|$ cannot be strictly less than $|u_i|$. This rules out case b).

We now derive the bounds on the value of $p$ as follows. Assume $p \geq 2q$, i.e., the degree of the log-pdf of each component of $U$ is at least twice the degree of the log-determinant of the Jacobian of $h$.

From the equation (23) we get the following bound on $p$

$$
\begin{aligned}
2c^p(d-1) &\leq \frac{1}{4} \\
\implies 8(d-1) &\leq \frac{1}{c^p} \\
\implies \log(8(d-1)) &\leq p\log(\frac{1}{c}) \\
\implies p &\geq \frac{\log(8(d-1))}{\log(\frac{1}{c})}
\end{aligned}
\tag{26}
$$

From the equation (24) we get the following bound on $p$

$$
\frac{a_{\mathsf{max}}(d-1)}{a_{\mathsf{min}}p} \leq \frac{1}{4} \implies p \geq \frac{4a_{\mathsf{max}}(d-1)}{a_{\mathsf{min}}}
\tag{27}
$$

From the equation (25) we get the following bound on $p$

$$\frac{b_{\max}}{|a_p|} \frac{n_{\text{poly}}}{(\alpha p^2)^{(p-q)}} \leq \frac{1}{4}$$

$$\implies \log\left(\frac{4b_{\max}n_{\text{poly}}}{|a_p|}\right) \leq 2(p-q)\log p \tag{28}$$

$$\implies p \geq \frac{\log\left(\frac{4b_{\max}n_{\text{poly}}}{a_{\min}}\right)}{2} + q$$

From the above equations (26), (27), and (28), we get that if

$$p \geq \max\left\{\frac{\log(8(d-1))}{\log(\frac{1}{c})}, \frac{4a_{\max}(d-1)}{a_{\min}}, \frac{\log\left(\frac{4b_{\max}n_{\text{poly}}}{a_{\min}}\right)}{2} + q\right\} \tag{29}$$

then the sum of the terms in the RHS in equation (22) is at most $\frac{3}{4}$ and the term in the LHS in equation (22) is 1, which leads to a contradiction.

From the above expression, we gather that the second and third term should dominate in determining the lower bound for $p$. From the second term, we gather that the lower bound increases linearly in the dimension of the latent, and from the third term we gather that $p$ must be greater than $q$ by a factor that grows logarithmically in the number of terms in the polynomial of the log determinant.

Let us now consider the case a) ( $|v_r| > |u_j|$) which is similar to the case b) analyzed above. Since values of $|u_i|$ are strictly less than $|u_j|$ and $|v_r|$, and since $|v_r| > |u_j|$, there exist a $c < 1$ such that $|u_i| \leq c\alpha p^2$, $|v_r| \geq \frac{1}{c}\alpha p^2$, $|v_i| \leq c|v_r|$, where $c < 1$. We follow the same steps as done in the analysis for case b). We separate the equation so that only the term $a_p$ (obtained by dividing $v_r^p$ with $v_r^p$) is in the LHS and the rest on the RHS.

$$|a_p| = \left|a_p \sum_{i=1}^{d-1}\left(\frac{u_i^p}{v_r^p} - \frac{v_i^p}{v_r^p}\right) + \sum_{k=1}^{p-1} a_k \sum_{i=1}^{d}\left(\frac{u_i^k}{v_r^p} - \frac{v_i^k}{v_r^p}\right) - \sum_m b_m \frac{1}{v_r^p}\prod_i u_i^{\theta_m(i)}\right|$$

$$\implies |a_p| \leq \left|a_p \sum_{i=1}^{d-1}\left(\frac{u_i^p}{v_r^p} - \frac{v_i^p}{v_r^p}\right)\right| + \left|\sum_{k=1}^{p-1} a_k \sum_{i=1}^{d}\left(\frac{u_i^k}{v_r^p} - \frac{v_i^k}{v_r^p}\right)\right| + \left|\sum_m b_m \frac{1}{u_j^p}\prod_i u_i^{\theta_m(i)}\right|$$

$$\implies 1 \leq \frac{1}{|a_p|}\left(\left|a_p \sum_{i=1}^{d-1}\left(\frac{u_i^p}{v_r^p} - \frac{v_i^p}{v_r^p}\right)\right| + \left|\sum_{k=1}^{p-1} a_k \sum_{i=1}^{d}\left(\frac{u_i^k}{v_r^p} - \frac{v_i^k}{v_r^p}\right)\right| + \left|\sum_m b_m \frac{1}{v_r^p}\prod_i u_i^{\theta_m(i)}\right|\right) \tag{30}$$

We analyze each of the terms in the RHS in equation (30) separately. The simplification of the first term in the RHS of the above yields the following upper bound.

$$\frac{1}{|a_p|}\left|a_p \sum_{i=1}^{d-1}\left(\frac{u_i^p}{v_r^p} - \frac{v_i^p}{v_r^p}\right)\right| \leq \frac{1}{|a_p|}|a_p| \sum_{i=1}^{d-1}\left(\left|\frac{u_i^p}{v_r^p}\right| + \left|\frac{v_i^p}{v_r^p}\right|\right) \tag{31}$$

$$\leq 2c^p(d-1)$$

The simplification of the second term in the RHS of equation (30) yields the following upper bound.

$$
\begin{aligned}
\frac{1}{|a_p|}\Big|\sum_{k=1}^{p-1}a_k\sum_{i=1}^{d}\Big(\frac{u_i^k}{v_r^p}-\frac{v_i^k}{v_r^p}\Big)\Big| &\leq \frac{1}{|a_p|}\sum_{k=1}^{p-1}|a_k|\sum_{i=1}^{d}\Big(\Big|\frac{u_i^k}{v_r^p}\Big|+\Big|\frac{v_i^k}{v_r^p}\Big|\Big)\Big| \\
&\leq \sum_{k=1}^{p-1}2|a_k|\sum_{i=1}^{d}\Big(\Big|\frac{u_j^k}{v_r^p}\Big|\Big)\Big| \\
&\leq \frac{1}{|a_p|}\Big(\sum_{k=1}^{p-1}2|a_k|(d-1)\Big)\frac{1}{\alpha p^2} \\
&\leq \frac{a_{\mathsf{max}}(d-1)c}{|a_p|\alpha p} \\
&\leq \frac{a_{\mathsf{max}}(d-1)}{a_{\mathsf{min}}p}
\end{aligned}
\tag{32}
$$

The simplification of the third term in the RHS of equation (30) yields the following upper bound.

$$
\begin{aligned}
\frac{1}{|a_p|}\Big|\sum_m b_m\frac{1}{v_r^p}\prod_i u_i^{\theta_m(i)}\Big| &\leq \sum_m|b_m|\frac{1}{|v_r|^{(p-q)}} \\
&\leq \frac{b_{\mathsf{max}}}{|a_p|}\frac{n_{\mathsf{poly}}\Big(c^{p-q}\Big)}{(\alpha p^2)^{(p-q)}} \\
&\leq \frac{b_{\mathsf{max}}}{|a_p|}\frac{n_{\mathsf{poly}}}{(\alpha p^2)^{(p-q)}}
\end{aligned}
\tag{33}
$$

Analyzing the RHS in equation (31)-(33), we see that if $p$ becomes sufficiently large then the RHS becomes less than 1. This contradicts the relationship in equation (30). Therefore, there is no $|v_i|$ that is strictly larger than $|u_i|$. This rules out case a). In fact from equations (31)-(33) we can get the same bound on $p$ as in case b).

Thus, the only possibility is case c), i.e., $|v_r| = |u_j| \implies v_r = u_j$ or $v_r = -u_j$. Consider the case when $p$ is odd. In that case, $v_r = u_j$ is the only option that works. We substitute $v_r = u_j$ in the equation (22) and repeat the same argument for the second highest absolute value, and so on. This leads to the conclusion that for each component $u$ there is a component of $v$ such that both of them are equal. Hence, we have established the relationship $v_r = u_j$. For another sample where index $j$ corresponds to the highest absolute value and is in the neighbourhood of $u_j$, the relationship $v_r = u_j$ must continue to hold. If this does not happen, then there would be another component $v_q = u_j$ where $q \neq r$. However, if that were the case, then it would contradict the continuity of $h$.

Therefore, the relationship $v_r = u_j$ (the match between index $j$ for $u$ and index $r$ for $v$) holds for a neighbourhood of values of vector $u$. Since each component of $h$ is analytic, we can use the fact that the neighbourhood of vector of values of $u$ for which the relationship $v_r = u_j$ holds has a positive measure, and then from (Mityagin, 2015) it follows that this relationship would hold on the entire space. We can draw the same conclusion for all the components of $h$ and conclude that $h$ is a permutation map.

∎

## Appendix D. Implementation Details

### D.1. Model Architecture

- Fully Connected Layer: (Data Dim, 100)

- BatchNormalization(100)

- LeakyReLU(0.5)

- Full Connected Layer: (100, Data Dim)

- BatchNormalization(Data Dim)

- LeakyReLU(0.5)

- Fully Connected Layer: (Data Dim, Total Tasks)

We consider the part of the network before the final fully connected layer as the representation network, and use the output from the representation network for training ICA/PCA after the ERM step.

### D.2. Hyperparameters

We use SGD to train all the methods with the different hyperparameters across each task mentioned below. In every case, we select the best model during the course of training based on the validation loss, and also use a learning rate scheduler that reduces the existing learning rate by half after every 50 epochs. Also, regarding ICA, we use the FastICA solver in sklearn with $30,000$ maximum iterations and data whitening.

- **Regression:** Learning Rate: $0.01$, Batch Size: $512$, Total Epochs: $1000$, Weight Decay: $5e-4$, Momentum: $0.9$

- **Classification:** Learning Rate: $0.05$, Batch Size: $512$, Total Epochs: $200$, Weight Decay: $5e-4$, Momentum: $0.9$

## Appendix E.  Additional Experiments

### E.1.  Task: Regression

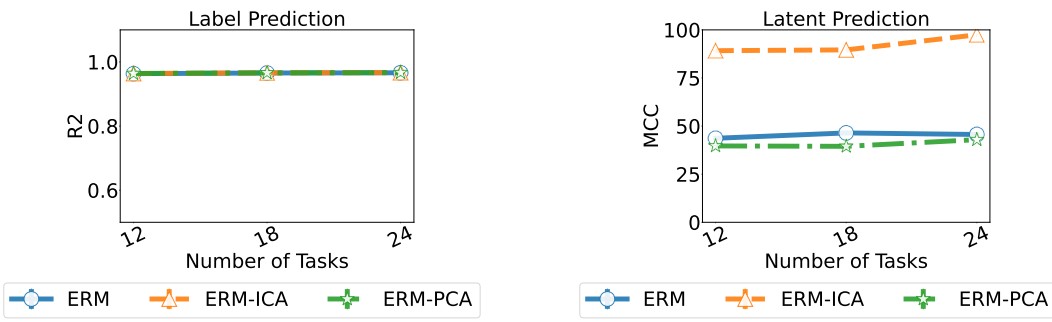

Figure 6: Regression Task: Data Dimension 24

### E.2.  Task: Classification

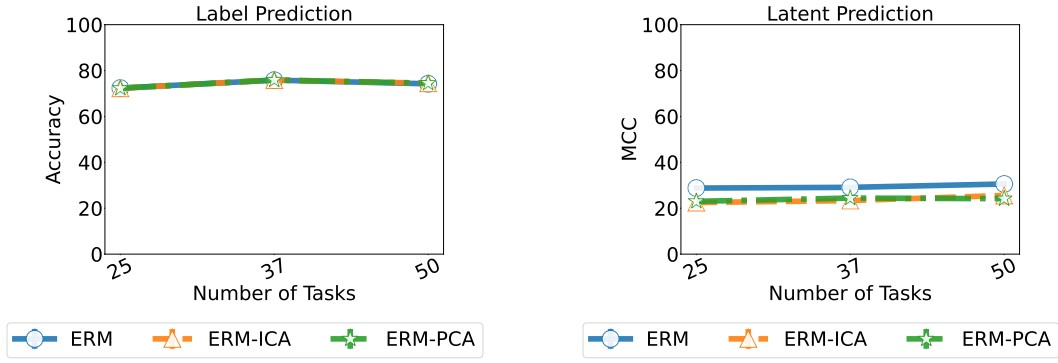

Figure 7: Classification Task: Data Dimension 50

