# OpenReview forum: "Towards efficient representation identification in supervised learning"
_cclear.cc/CLeaR/2022/Conference — CLeaR 2022 Poster_

### Official Review · Reviewer_xwJZ · 2021-11-22

**Confidence:** 2
**Overall Score:** 6

**Main Review:**

This paper is well written and easy to understand. I think the idea of constraining representations with i.i.d assumption would bring insights to the community.

1. It would be interesting to see some label-generate-latent rivals in the experiments.
2. typos in the Proof sketch of Theorem 3: 'Baseed' -> 'Based'

**Summary:**

This paper extends ERM paradigm with ICA under the setting of latents-generate-label process. Based on the assumption that all components of the representation layer are mutually independent, the authors provide proofs and experimental evidences to show the effectiveness of their method.

---

> ### Author Response · Authors · 2021-12-03
> **Different data generation process in our work and previous works**
>
> We thank the reviewer for their encouraging feedback! We provide clarification regarding the questions raised by them below.
>
> * We highlight that our data generation process assumes that the latent variables cause the labels. Hence, we cannot compare with the approaches that work under the assumption that the labels cause the latent variables as it would not be a fair comparison. Our approach and the previous approaches are not directly comparable, hence based on the underlying data generation process we should decide which approach is suitable for identifying the latent variables. Inferring what model explains the data better (latent variables  $\rightarrow$ labels versus labels $\rightarrow$ latent variables), followed by identification using the right approach is an important problem to be addressed in a separate work.
>
> * We will correct the typo in Theorem 3 and any other typos in the paper.

---

### Official Review · Reviewer_N1Q2 · 2021-11-24

**Confidence:** 3
**Overall Score:** 7

**Main Review:**

The Nonlinear ICA problem is interesting. The authors investigate this problem in a natural setting, which is different from the previous. The theoretical result is complete and covers many situations. The experimental results confirm the effectiveness of the proposed ERM-ICA.
There are some confusing statement.
In Page 8, I think "If the predictor is \theta^{\top}\Phi(X)=1^{\top}(\Phi(X)\odot\frac{1}{\theta})" should be "If the predictor is \theta^{\top}\Phi(X)=1^{\top}(\Phi(X)\odot\theta)".
In Definition 2, I think the \Phi(X) should be V.

**Summary:**

This paper investigate the Nolinear Independent Component Analysis problem in a new setting, where latent factors cause auxiliary information, rather than auxiliary information cause latent factors. The authors prove that the latent factors can be recovered up to permutation and scaling with some excess assumption. When the dimension of label is at least equal to the dimension of factors, the assumption is weak. When the dimension of label is one, the assumption is stronger. To learn the latent factors from the observations and labels, the authors propose IC-ERM which constrains the representaion layer are mutually independent. To implement it,  the authors propose ERM-ICA.

---

> ### Author Response · Authors · 2021-12-02
> **Correcting the typos in equation and definition**
>
> We thank the reviewer for their encouraging feedback! We provide clarifications about the specific confusing statements raised by the reviewer and will update the paper later regarding any typos in general.
>
> - We will correct the equation on page 8 to component wise mutiplication of $\Phi(X)$ and $\theta$; $\Phi(X) \odot \theta =  [\theta_1*\Phi_1(X),  \cdots, \theta_d * \Phi_d(X)] $
>
> - We will update the typo in Definition 2, it should be $V$ instead of $\Phi(X)$.

---

### Official Review · Reviewer_Y9XU · 2021-11-24

**Confidence:** 3
**Overall Score:** 6

**Main Review:**

Overall, the paper achieves the identification goal by placing somewhat different (but possibly strong) conditions on the data generating process, so the identification becomes somewhat easier. Nonetheless, it should be a good addition to the current literature.

I have some minor questions regarding the definition of *tasks*.
-	In particular, what does it mean by “different tasks” and how they would affect the identity proof? For instance, if some tasks are redundant, then $k=d$ tasks may not be enough. There should be an explicit condition on how diverse the tasks are.
-	In the first two results, the theorem says the number of tasks is exactly equal to the number of latent confounders, i.e., $k=d$. What if $k>d$? How do we know the condition in practice? And how do we select $d$ in practical applications?
.


**Summary:**

The paper considers representation identification in the setting where latent factors generate observations and labels for multiple tasks, and the latent factors are mutually independent. Within somewhat additional assumptions (compared with existing works), authors show that disentanglement (up to permutation and scaling) is possible in theory and also by experiment.

---

> ### Author Response · Authors · 2021-12-03
> **Clarification regarding the relationship between k and d**
>
> We thank the reviewer for their encouraging feedback! We address the concerns raised by them below.
>
> - **Task Diversity:** The diversity of tasks is captured by the constraint that $\Gamma$ is invertible (Assumption 2). Hence, we cannot have redundant tasks for the identification of the latent variables. However, we could have a poorly conditioned invertible matrix, which would lead to slower numerical optimization in linear ICA. We will update our paper with a discussion regarding this.
>
> - **k > d scenario:** When the number of tasks are greater than the data dimension ($k > d$), we could select a subset $S^{'}$ of tasks such that $|S^{'}|=d$. Then we can proceed ahead in a similar fashion as Theorem 1. This question arises commonly in linear ICA literature; selecting a subset of tasks is standard practice. There are other ways to address this issue as well that involve PCA, we will also elaborate on that in the paper. Also, we had shown in our paper (Theorem 3) that we can identify the latent variables when $k=1$ under certain conditions, which implies in general the case of $k<d$ can be handled. Hence, we aim to be agnostic to the exact relationship between $k$ and $d$.
>
> - **Selecting data dimension d:** Our approach comprises of two phases, first training under the ERM objective, followed by linear ICA on the representation. So we can treat $d$ as a hyperparameter and tune it based on the linear ICA objective value used in the latter part of our procedure. Although assuming the value of $d$ is known is a common practice in previous works (Zimmerman et al.), we thank the reviewer for this suggestion and would explore this further with our validation strategy proposed above.

---

### Decision · Program_Chairs · 2022-01-13

**Decision:**

Accept (Poster)

**Comment:**

I follow the recommendation of the three reviewers to accept this paper. I encourage the authors to address the questions and typos of the reviewers.